# Rethinking Pre-training and Self-training

**Barret Zoph**,* **Golnaz Ghiasi**,* **Tsung-Yi Lin**,*
**Yin Cui, Hanxiao Liu, Ekin D. Cubuk, Quoc V. Le**
Google Research, Brain Team
{barretzoph,golnazg,tsungyi,yincui,hanxiaol,cubuk,qvl}@google.com

## Abstract

Pre-training is a dominant paradigm in computer vision. For example, supervised ImageNet pre-training is commonly used to initialize the backbones of object detection and segmentation models. He et al. [1], for example, show a contrasting result that ImageNet pre-training has limited impact on COCO object detection. Here we investigate self-training as another method to utilize additional data on the same setup and contrast it against ImageNet pre-training. Our study reveals the generality and flexibility of self-training with three additional insights: 1) stronger data augmentation and more labeled data further diminish the value of pre-training, 2) unlike pre-training, self-training is always helpful when using stronger data augmentation, in both low-data and high-data regimes, and 3) in the case that pre-training is helpful, self-training improves upon pre-training. For example, on the COCO object detection dataset, pre-training benefits when we use one fifth of the labeled data, and *hurts* accuracy when we use all labeled data. Self-training, on the other hand, shows positive improvements from +1.3 to +3.4AP across all dataset sizes. In other words, self-training works well exactly on the same setup that pre-training does not work (using ImageNet to help COCO). On the PASCAL segmentation dataset, which is a much smaller dataset than COCO, though pre-training does help significantly, self-training improves upon the pre-trained model. On COCO object detection, we achieve 54.3AP, an improvement of +1.5AP over the strongest SpineNet model. On PASCAL segmentation, we achieve 90.5 mIOU, an improvement of +1.5% mIOU over the previous state-of-the-art result by DeepLabv3+.[1]

## 1 Introduction

Pre-training is a dominant paradigm in computer vision. As many vision tasks are related, it is expected a model, pre-trained on one dataset, to help another. It is now common practice to pre-train the backbones of object detection and segmentation models on ImageNet classification [2–5]. This practice has been recently challenged He et al. [1], among others [6,7], who show a surprising result that such ImageNet pre-training does not improve accuracy on the COCO dataset.

A stark contrast to pre-training is self-training [8–10]. Let's suppose we want to use ImageNet to help COCO object detection; under self-training, we will first discard the labels on ImageNet. We then train an object detection model on COCO, and use it to generate pseudo labels on ImageNet. The pseudo-labeled ImageNet and labeled COCO data are then combined to train a new model. The recent successes of self-training [11–14] raise the question to what degree does self-training work better than pre-training. Can self-training work well on the exact setup, using ImageNet to improve COCO, where pre-training fails?

Our work studies self-training with a focus on answering the above question. We define a set of control experiments where we use ImageNet as additional data with the goal of improving COCO. We vary the amount of labeled data in COCO and the strength of data augmentation as control factors. Our experiments show that as we increase the strength of data augmentation or the amount of labeled data, the value of pre-training diminishes. In fact, with our strongest data augmentation, pre-training significantly *hurts* accuracy by -1.0AP, a surprising result that was not seen by He et al. [1]. Our experiments then show that self-training interacts well with data augmentations: stronger data augmentation not only doesn't hurt self-training, but also helps it. Under the same data augmentation, self-training yields positive +1.3AP improvements using the same ImageNet dataset. This is another striking result because it shows self-training works well exactly on the setup that pre-training fails. These two results provide a positive answer to the above question.

An increasingly popular pre-training method is self-supervised learning. Self-supervised learning methods pre-train on a dataset without using labels with the hope to build more universal representations that work across a wider variety of tasks and datasets. We study ImageNet models pre-trained using a state-of-the-art self-supervised learning technique and compare to standard supervised ImageNet pre-training on COCO. We find that self-supervised pre-trained models using SimCLR [15] have similar performance as supervised ImageNet pre-training. Both methods *hurt* COCO performance in the high data/strong augmentation setting, when self-training helps. Our results suggest that both supervised and self-supervised pre-training methods fail to scale as the labeled dataset size grows, while self-training is still useful.

Our work however does not dismiss the use of pre-training in computer vision. Fine-tuning a pre-trained model is faster than training from scratch and self-training in our experiments. The speedup ranges from 1.3x to 8x depending on the pre-trained model quality, strength of data augmentation, and dataset size. Pre-training can also benefit applications where collecting sufficient labeled data is difficult. In such scenarios, pre-training works well; but self-training also benefits models with and without pre-training. For example, our experiment with PASCAL segmentation dataset shows that ImageNet pre-training improves accuracy, but self-training provides an additional +1.3% mIOU boost on top of pre-training. The fact that the benefit of pre-training does not cancel out the gain by self-training, even when utilizing the same dataset, suggests the generality of self-training.

Taking a step further, we explore the limits of self-training on COCO and PASCAL datasets, thereby demonstrating the method's flexibility. We perform self-training on COCO dataset with Open Images dataset as the source of unlabeled data, and RetinaNet [16] with SpineNet [17] as the object detector. This combination achieves 54.3AP on the COCO test set, which is +1.5AP better than the strongest SpineNet model. On segmentation, we use PASCAL aug set [18] as the source of unlabeled data, and NAS-FPN [19] with EfficientNet-L2 [12] as the segmentation model. This combination achieves 90.5AP on the PASCAL VOC 2012 test set, which surpasses the state-of-the-art accuracy of 89.0AP [20], who also use additional 300M labeled images. These results confirm another benefit of self-training: it's very flexible about unlabeled data sources, model architectures and computer vision tasks.

## 2    Related Work

Pre-training has received much attention throughout the history of deep learning (see [21] and references therein). The resurgence of deep learning in the 2000s also began with unsupervised pre-training [22–26]. The success of unsupervised pre-training in NLP [27–32] has revived much interest in unsupervised pre-training in computer vision, especially contrastive training [15, 33–37]. In practice, supervised pre-training is highly successful in computer vision. For example, many studies, e.g., [38–42], have shown that ConvNets pre-trained on ImageNet, Instagram, and JFT can provide strong improvements for many downstream tasks.

Supervised ImageNet pre-training is the most widely-used initialization method for machine vision (e.g., [2–5]). Shen et al [6] and He et al. [1], however, demonstrate that ImageNet pre-training does not work well if we consider a much different task such as COCO object detection. Ghiasi et al. [7] find model trained with random initialization outperforms the ImageNet pre-trained model on COCO object detection when strong regularization is applied. Poudel et al [43] on the other hand show that ImageNet pre-training is not necessary for semantic segmentation with CityScapes if aggressive data augmentation is applied. Furthermore, Raghu et al. [44] show that ImageNet pre-training does not

improve medical image classification tasks. Compared to these previous works, our work takes a step further and studies the role of pre-training in computer vision in greater detail with stronger data augmentation, different pre-training methods (supervised and self-supervised), and different pre-trained checkpoint qualities.

Our paper does not study targeted pre-training in depth, e.g., using an object detection dataset to improve another object detection dataset, for two reasons. Firstly, targeted pre-training is expensive and not scalable. Secondly, there exists evidence that pre-training on a dataset that is the same as the target task still can fail to yield improvements. For example, Shao et al. [45] found that pre-training on the Open Images object detection dataset actually *hurts* COCO performance. More analysis of targeted pre-training can be found in [46].

Our work argues for the scalability and generality of self-training (e.g., [8–10]). Recently, self-training has shown significant progress in deep learning (e.g., image classification [11,12], machine translation [13], and speech recognition [14,47]). Most closely related to our work is Xie et al. [12] who also use strong data augmentation in self-training but for image classification. Closer in applications are semi-supervised learning for detection and segmentation (e.g., [48–52]), who only study self-training in isolation or without a comparison against ImageNet pre-training. They also do not consider the interactions between these training methods and data augmentations.

## 3 Methodology

### 3.1 Methods and Control Factors

**Data Augmentation:**   We use four different augmentation policies of increasing strength that work for both detection and segmentation. This allows for varying the strength of data augmentation in our analysis. We design our augmentation policies based on the standard flip and crop augmentation in the literature [16], AutoAugment [53,54], and RandAugment [55]. The standard flip and crop policy consists of horizontal flips and scale jittering [16]. The random jittering operation resizes an image to (0.8, 1.2) of the target image size and then crops it. AutoAugment and RandAugment are originally designed with the standard scale jittering. We increase scale jittering (0.5, 2.0) in AutoAugment and RandAugment and find the performances are significantly improved. For RandAugment we use a magntiude of 10 for all models [55]. We arrive at our four data augmentation policies which we use for experimentation: FlipCrop, AutoAugment, AutoAugment with higher scale jittering, RandAugment with higher scale jittering. Throughout the text we will refer to them as: **Augment-S1**, **Augment-S2**, **Augment-S3** and **Augment-S4** respectively. The last three augmentation policies are stronger than He et al. [1] who use only a FlipCrop-based strategy.

**Pre-training:**   To evaluate the effectiveness of pre-training, we study ImageNet pre-trained checkpoints of varying quality. To control for model capacity, all checkpoints use the same model architecture but have different accuracies on ImageNet (as they were trained differently). We use the EfficientNet-B7 architecture [56] as a strong baseline for pre-training. For the EfficientNet-B7 architecture, there are two available checkpoints: 1) the EfficientNet-B7 checkpoint trained with AutoAugment that achieves 84.5% top-1 accuracy on ImageNet; 2) the EfficientNet-B7 checkpoint trained with the Noisy Student method [12], which utilizes an additional 300M unlabeled images, that achieves 86.9% top-1 accuracy.[2] We denote these two checkpoints as **ImageNet** and **ImageNet++** , respectively. Training from a random initialization is denoted by **Rand Init**. All of our baselines are therefore stronger than He et al. [1] who only use ResNets for their experimentation (EfficientNet-B7 checkpoint has an approximately 8% higher accuracy than a ResNet-50 checkpoint). Table 1 summarizes our notations for data augmentations and pre-trained checkpoints.

**Self-training:**   We use a simple self-training method inspired by [9,12,48,57] which consists of three steps. First, a teacher model is trained on the labeled data (e.g., COCO dataset). Then the teacher model generates pseudo labels on unlabeled data (e.g., ImageNet dataset). Finally, a student is trained to optimize the loss on human labels and pseudo labels jointly. Our experiments with various hyperparameters and data augmentations indicate that self-training with this standard loss function can be unstable. To address this problem, we implement a loss normalization technique, which is described in Appendix B.

| Name | Description |
|---|---|
| **Augment-S1** | Weakest augmentation: Flips and Crops |
| **Augment-S2** | Third strongest augmentation: AutoAugment, Flips and Crops |
| **Augment-S3** | Second strongest augmentation: Large Scale Jittering, AutoAugment, Flips and Crops |
| **Augment-S4** | Strongest augmentation: Large Scale Jittering, RandAugment, Flips and Crops |
| **Rand Init** | Model initialized w/ random weights |
| **ImageNet Init** | Model initialized w/ ImageNet pre-trained checkpoint (84.5% top-1) |
| **ImageNet++ Init** | Model initialized w/ higher performing ImageNet pre-trained checkpoint (86.9% top-1) |

Table 1: Notations for data augmentations and pre-trained models used throughout this work.

## 3.2 Additional Experimental Settings

**Object Detection:** We use COCO dataset [58] (118k images) for supervised learning. In self-training, we experiment with ImageNet [59] (1.2M images) and OpenImages [60] (1.7M images) as unlabeled datasets. We adopt RetinaNet detector [16] with EfficientNet-B7 backbone and feature pyramid networks [61] in the experiments. We use image size $640 \times 640$, pyramid levels from $P_3$ to $P_7$ and 9 anchors per pixel as done in [16]. The training batch size is 256 with weight decay 1e-4. The model is trained with learning rate 0.32 and a cosine learning rate decay schedule [62]. At the beginning of training the learning rate is linearly increased over the first 1000 steps from 0.0032 to 0.32. All models are trained using synchronous Batch Normalization. For all experiments using different augmentation strengths and datasets sizes, we allow each model to train until it converges (when training longer stops helping or even hurts performance on a held-out validation set). For example, training takes 45k iterations with Augment-S1 and 120k iterations with Augment-S4, when both models are randomly initialized. For results using SpineNet, we use the model architecture and hyper-parameters reported in the paper [17]. When we use SpineNet, due to memory constraints we reduce the batch size from 256 to 128 and scale the learning rate by half. The hyper-parameters, except batch size and learning rate, follow the default implementation in the SpineNet open-source repository.[3] All SpineNet models also use Soft-NMS with a sigma of 0.3 [63]. In self-training, we use a hard score threshold of 0.5 to generate pseudo box labels. We use a total 512 batch size with 256 from COCO and 256 from a pseudo dataset. The other training hyper-parameters remain the same as those in supervised training. For all experiments the parameters of the student model are initialized by the teacher model to save training time. Experimental analysis studying the impact of student model initialization during self-training can be found in Appendix C.

**Semantic Segmentation:** We use the `train` set (1.5k images) of PASCAL VOC 2012 segmentation dataset [64] for supervised learning. In self-training, we experiment with augmented PASCAL dataset [18] (9k images), COCO [58] (240k images, combining labeled and unlabeled datasets), and ImageNet [59] (1.2M images). We adopt a NAS-FPN [19] model architecture with EfficientNet-B7 and EfficientNet-L2 backbone models. Our NAS-FPN model uses 7 repeats with depth-wise separable convolution. We use pyramid levels from $P_3$ to $P_7$ and upsample all feature levels to $P_2$ and then merge them by a sum operation. We apply 3 layers of $3 \times 3$ convolutions after the merged features and then attach a $1 \times 1$ convolution for 21 class prediction. The learning rate is set to 0.08 for EfficientNet-B7 and 0.2 for EfficientNet-L2 with batch size 256 and weight decay 1e-5. All models are trained with a cosine learning rate decay schedule and use synchronous Batch Normalization. EfficientNet-B7 is trained for 40k iterations and EfficientNet-L2 for 20k iterations. For self-training, we use a batch size of 512 for EfficientNet-B7 and 256 for EfficientNet-L2. Half of the batch consists of supervised data and the other half pseudo data. Other hyper-parameters follow those in the supervised training. Additionally, we use a hard score threshold of 0.5 to generate segmentation masks and pixels with a smaller score are set to the ignore label. Lastly, we apply multi-scale inference augmentation with scales of (0.5, 0.75, 1, 1.25, 1.5, 1.75) to compute the segmentation masks for pseudo labeling.

In Appendix H, we show the optimal training iterations and loss hyperparameters used for all of our experiments.

# 4  Experiments

## 4.1  The effects of augmentation and labeled dataset size on pre-training

This section expands on the findings of He et al. [1] who study the weaknesses of pre-training on the COCO dataset as they vary the size of the labeled dataset. Similar to their study, we use ImageNet for supervised pre-training and vary the COCO labeled dataset size. Different from their study, we also change other factors: data augmentation strengths and pre-trained model qualities (see Section 3.1 for more details). As mentioned above, we use RetinaNet object detectors with the EfficientNet-B7 architecture as the backbone. Below are our key findings:

**Pre-training hurts performance when stronger data augmentation is used.**  We analyze the impact of pre-training when we vary the augmentation strength. In Figure 1-Left, when we use the standard data augmentation (Augment-S1), pre-training helps. But as we increase the data augmentation strength, the value of pre-training diminishes.

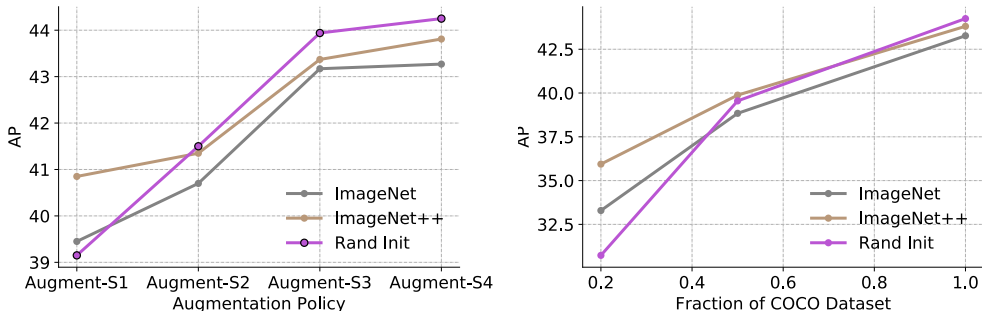

Figure 1: The effects of data augmentation and dataset size on pre-training. **Left**: Supervised object detection performance under various ImageNet pre-trained checkpoint qualities and data augmentation strengths on COCO. **Right**: Supervised object detection performance under various COCO dataset sizes and ImageNet pre-trained checkpoint qualities. All models use Augment-S4 (for similar results with other augmentation methods see Appendix D).

Furthermore, in the stronger augmentation regimes, we observe that pre-training actually *hurts* performance by a large amount (-1.0 AP). This result was not observed by He et al. [1], as pre-training only slightly hurts performance (-0.4AP) or is neutral in their experiments.

**More labeled data diminishes the value of pre-training.**  Next, we analyze the impact of pre-training when varying the labeled dataset size. Figure 1-Right shows that pre-training is helpful in the low-data regimes (20%) and neutral or harmful in the high-data regime. This result is mostly consistent with the observation in He et al. [1]. One new finding here is that the checkpoint quality does correlate with the final performance in the low data regime (ImageNet++ performs best on 20% COCO).

## 4.2  The effects of augmentation and labeled dataset size on self-training

In this section, we analyze self-training and contrast it with the above results. For consistency, we will continue to use COCO object detection as the task of interest, and ImageNet as the self-training data source. Unlike pre-training, self-training only treats ImageNet as unlabeled data. Again, we use RetinaNet object detectors with the EfficientNet-B7 architecture as the backbone to be compatible with previous experiments. Below are our key findings:

**Self-training helps in high data/strong augmentation regimes, even when pre-training hurts.** Similar to the previous section, we first analyze the performance of object detectors as we vary the data augmentation strength. Table 2 shows the performance of self-training across the four data augmentation methods, and compares them against supervised learning (Rand Init) and pre-training (ImageNet Init). Here we also show the gain/loss of self-training and pre-training to the baseline. The results confirm that in the scenario where pre-training hurts (strong data augmentations: Augment-S2,

Augment-S3, Augment-S4), self-training helps significantly. It provides a boost of more than +1.3AP on top of the baseline, when pre-training hurts by -1.0AP. Similar results are obtained on ResNet-101 (see Appendix E).

| Setup | Augment-S1 | Augment-S2 | Augment-S3 | Augment-S4 |
|---|---|---|---|---|
| Rand Init | 39.2 | 41.5 | 43.9 | 44.3 |
| ImageNet Init | (+0.3) 39.5 | (-0.7) 40.7 | (-0.8) 43.2 | (-1.0) 43.3 |
| Rand Init w/ ImageNet Self-training | (+1.7) 40.9 | (+1.5) 43.0 | (+1.5) 45.4 | (+1.3) 45.6 |

Table 2: In regimes where pre-training hurts, self-training with the same data source helps. All models are trained on the full COCO dataset.

**Self-training works across dataset sizes and is additive to pre-training.** Next we analyze the performance of self-training as we vary the COCO labeled dataset size. As can be seen from Table 3, self-training benefits object detectors across dataset sizes, from small to large, regardless of pre-training methods. Most importantly, at the high data regime of 100% labeled set size, self-training significantly improves all models while pre-training hurts.

In the low data regime of 20%, self-training enjoys the biggest gain of +3.4AP on top of Rand Init. This gain is bigger than the gain achieved by ImageNet Init (+2.6AP). Although the self-training gain is smaller than the gain by ImageNet++ Init, ImageNet++ Init uses 300M additional unlabeled images.

Self-training is quite additive with pre-training even when using the *same data source*. For example, in the 20% data regime, utilizing an ImageNet pre-trained checkpoint yields a +2.6AP boost. Using both pre-training and self-training with ImageNet yields an additional +2.7AP gain. The additive benefit of combining pre-training and self-training is observed across all of the dataset sizes.

| Setup | 20% Dataset | 50% Dataset | 100% Dataset |
|---|---|---|---|
| Rand Init | 30.7 | 39.6 | 44.3 |
| Rand Init w/ ImageNet Self-training | (+3.4) 34.1 | (+1.8) 41.4 | (+1.3) 45.6 |
| ImageNet Init | 33.3 | 38.8 | 43.3 |
| ImageNet Init w/ ImageNet Self-training | (+2.7) 36.0 | (+1.7) 40.5 | (+1.3) 44.6 |
| ImageNet++ Init | 35.9 | 39.9 | 43.8 |
| ImageNet++ Init w/ ImageNet Self-training | (+1.3) 37.2 | (+1.6) 41.5 | (+0.8) 44.6 |

Table 3: Self-training improves performance for all model initializations across all labeled dataset sizes. All models are trained on COCO using Augment-S4.

### 4.3 Self-supervised pre-training also hurts when self-training helps in high data/strong augmentation regimes

The previous experiments show that ImageNet pre-training hurts accuracy, especially in the highest data and strongest augmentation regime. Under this regime, we investigate another popular pre-training method: self-supervised learning.

The primary goal of self-supervised learning, pre-training without labels, is to build universal representations that are transferable to a wider variety of tasks and datasets. Since supervised ImageNet pre-training hurts COCO performance, potentially self-supervised learning techniques not using label information could help. In this section, we focus on COCO in the highest data (100% COCO dataset) and strongest augmentation (Augment-S4) setting. Our goal is to compare random initialization against a model pre-trained with a state-of-the-art self-supervised algorithm. For this purpose, we choose a checkpoint that is pre-trained with the SimCLR framework [15] on ImageNet. We use the checkpoint before it is fine-tuned on ImageNet labels. All backbones models use ResNet-50 as SimCLR only uses ResNets in their work.

The results in Table 4 reveal that the self-supervised pre-trained checkpoint hurts performance just as much as supervised pre-training on the COCO dataset. Both pre-trained models *decrease* performance by -0.7AP over using a randomly initialized model. Once again we see self-training improving performance by +0.8AP when both pre-trained models hurt performance. Even though both self-supervised learning and self-training ignore the labels, self-training seems to be more effective at using the unlabeled ImageNet data to help COCO.

| Setup | COCO AP |
|---|---|
| Rand Init | 41.1 |
| ImageNet Init (Supervised) | (-0.7) 40.4 |
| ImageNet Init (SimCLR) | (-0.7) 40.4 |
| Rand Init w/ Self-training | (+0.8) 41.9 |

Table 4: Self-supervised pre-training (SimCLR) hurts performance on COCO just like standard supervised pre-training. Performance of **ResNet-50** backbone model with different model initializations on full COCO. All models use Augment-S4.

## 4.4 Exploring the limits of self-training and pre-training

In this section we combine our knowledge about the interactions of data augmentation, self-training and pre-training to improve the state-of-the-art. Below are our key results:

**COCO Object Detection.** In this experiment, we use self-training and Augment-S3 as the augmentation method. The previous experiments on full COCO suggest that ImageNet pre-training hurts performance, so we do not use it. Although the control experiments use EfficientNet and ResNet backbones, we use SpineNet [17] in this experiment as it is closer to the state-of-the-art. For self-training, we use Open Images Dataset (OID) [60] as the self-training unlabeled data, which we found to be better than ImageNet (for more information about the effects of data sources on self-training, see Appendix F). Note that OID is found to not be helpful on COCO by pre-training in [45].

Table 5 shows our results on the two largest SpineNet models, and compares them against previous best single-model, single-crop performance on this dataset. For the largest SpineNet model we improve upon the best 52.8AP SpineNet model by +1.5AP to achieve 54.3AP. Across all model variants, we obtain at least a +1.5AP gain.

| Model | # FLOPs | # Params | AP (`val`) | AP (`test-dev`) |
|---|---|---|---|---|
| AmoebaNet+ NAS-FPN+AA (1536) | 3045B | 209M | 50.7 | — |
| EfficientDet-D7 (1536) | 325B | 52M | 52.1 | 52.6 |
| SpineNet-143[†] (1280) | 524B | 67M | 50.9 | 51.0 |
| SpineNet-143 (1280) w/ Self-training | 524B | 67M | (+1.5) **52.4** | (+1.6) **52.6** |
| SpineNet-190[†] (1280) | 1885B | 164M | 52.6 | 52.8 |
| SpineNet-190 (1280) w/ Self-training | 1885B | 164M | (+1.6) **54.2** | (+1.5) **54.3** |

Table 5: Comparison with the strong models on COCO object detection. Self-training results use the Open Images Dataset. Parentheses next to the model name denote the training image size. [†] The SpineNet baselines here do not contains the **Augment-S3** that is used in the Self-training experiments as the models were found to be too unstable and were unable to finish training.

**PASCAL VOC Semantic Segmentation.** For this experiment, we use NAS-FPN architecture [19] with EfficientNet-B7 [56] and EfficientNet-L2 [12] as the backbone architectures. Due to PASCAL's small dataset size, pre-training still matters much here. Hence, we use a combination of pre-training, self-training and strong data augmentation for this experiment. For pre-training, we use the ImageNet++ for the EfficientNet backbones. For augmentation, we use Augment-S4. We use the `aug` set of PASCAL [18] as the additional data source for self-training, which we found to be more effective than ImageNet.

Table 6 shows that our method improves state-of-the-art by a large margin. We achieve 90.5% mIOU on the PASCAL VOC 2012 test set using single-scale inference, outperforming the old state-of-the-art 89% mIOU which utilizes multi-scale inference. For PASCAL, we find pre-training with a good checkpoint to be crucial, without it we achieve 41.5 % mIOU. Interestingly, our model improves the previous state-of-the-art by 1.5% mIOU even using much less human labels in training. Our method uses labeled data in ImageNet (1.2M images) and PASCAL train segmentation (1.5k images). In contrast, previous state-of-the-art models [65] used 250x additional pre-training labeled classification data: JFT (300M images), and 86x additional labeled segmentation data: COCO (120k images), and PASCAL `aug` (9k images). For a visualization of pseudo labeled images, see Appendix G.

| Model | Pre-trained | # FLOPs | # Params | mIOU (`val`) | mIOU (`test`) |
|---|---|---|---|---|---|
| ExFuse [†] | ImageNet, COCO | | | 85.8 | 87.9 [‡] |
| DeepLabv3+ | ImageNet | 177B | | 80.0 | — |
| DeepLabv3+ | ImageNet, JFT, COCO | 177B | | 83.4 | — |
| DeepLabv3+ [†] | ImageNet, JFT, COCO | 3055B | | 84.6 | 89.0 [‡] |
| Eff-B7 | ImageNet++ | 60B | 71M | 85.2 | — |
| Eff-B7 w/ Self-training | ImageNet++ | 60B | 71M | (+1.5) **86.7** | — |
| Eff-L2 | ImageNet++ | 229B | 485M | 88.7 | — |
| Eff-L2 w/ Self-training | ImageNet++ | 229B | 485M | (+1.3) **90.0** | **90.5** |

Table 6: Comparison with state-of-the-art models on PASCAL VOC 2012 `val`/`test` set. [†] indicates multi-scale/flip ensembling inference. [‡] indicates fine tuning the model on the `train+val` with hard classes being duplicated [20]. EfficientNet models (Eff) are trained on PASCAL `train` set for validation results and `train+val` for test results. Self-training uses the `aug` set of PASCAL.

## 5 Discussion

**Rethinking pre-training and universal feature representations.** One of the grandest goals of computer vision is to develop universal feature representations that can solve many tasks. Our experiments show the limitation of learning universal representations from both classification and self-supervised tasks, demonstrated by the performance differences in self-training and pre-training. Our intuition for the weak performance of pre-training is that pre-training is not aware of the task of interest and can fail to adapt. Such adaptation is often needed when switching tasks because, for example, good features for ImageNet may discard positional information which is needed for COCO. We argue that jointly training the self-training objective with supervised learning is more adaptive to the task of interest. We suspect that this leads self-training to be more generally beneficial.

**The benefit of joint-training.** A strength of the self-training paradigm is that it jointly trains the supervised and self-training objectives, thereby addressing the mismatch between them. But perhaps we can jointly train ImageNet and COCO to address this mismatch too? Table 7 shows results for *joint-training*, where ImageNet classification is trained jointly with COCO object detection (we use the exact setup as self-training in this experiment). Our results indicate that ImageNet pre-training yields a +2.6AP improvement, but using a random initialization and joint-training gives a comparable gain of +2.9AP. This improvement is achieved by training *19 epochs* over the ImageNet dataset. Most ImageNet models that are used for fine-tuning require much longer training. For example, the ImageNet Init (supervised pre-trained model) needed to be trained for 350 epochs on the ImageNet dataset.

Moreover, pre-training, joint-training and self-training are all additive using the same ImageNet data source (last column of the table). ImageNet pre-training gets a +2.6AP improvement, pre-training + joint-training gets +0.7AP improvement and doing pre-training + joint-training + self-training achieves a +3.3AP improvement.

| Setup | Sup. Training | w/ Self-training | w/ Joint Training | w/ Self-training w/ Joint Training |
|---|---|---|---|---|
| Rand Init | 30.7 | (+3.4) 34.1 | (+2.9) 33.6 | (+4.4) 35.1 |
| ImageNet Init | 33.3 | (+2.7) 36.0 | (+0.7) 34.0 | (+3.3) 36.6 |

Table 7: Comparison of pre-training, self-training and joint-training on COCO. All three methods use ImageNet as the additional dataset source. All models are trained on 20% COCO with Augment-S4.

**The importance of task alignment.** One interesting result in our experiments is ImageNet pre-training, even with additional human labels, performs worse than self-training. Similarly, we verify the same phenomenon on PASCAL dataset. On PASCAL dataset, the `aug` set is often used as an additional dataset, which has much noisier labels than the `train` set. Our experiment shows that with strong data augmentation (Augment-S4), training with `train+aug` actually hurts accuracy. Meanwhile, pseudo labels generated by self-training on the same `aug` dataset significantly improves accuracy. Both results suggest that noisy (PASCAL) or un-targeted (ImageNet) labeling is worse than targeted pseudo labeling.

It is worth mentioning that Shao et al. [45] report pre-training on Open Images hurts performance on COCO, despite both of them being annotated with bounding boxes. This means that not only

| Setup | train | train + aug | train + aug w/ Self-training |
|---|---|---|---|
| ImageNet Init w/ Augment-S1 | 83.9 | (+0.8) 84.7 | (+1.7) 85.6 |
| ImageNet Init w/ Augment-S4 | 85.2 | (-0.4) 84.8 | (+1.5) 86.7 |

Table 8: Performance on PASCAL VOC 2012 using `train` or `train` and `aug` for the labeled data. Training on `train + aug` hurts performance when strong augmentation (Augment-S4) is used, but training on `train` while utilizing `aug` for self-training improves performance.

we want the task to be the same but also the annotations to be the same for pre-training to be really beneficial. Self-training on the other hand is very general and can use Open Images successfully to improve COCO performance in Appendix F, a result that suggests self-training can align to the task of interest well.

**Limitations.** There are still limitations to current self-training techniques. In particular, self-training requires more compute than fine-tuning on a pre-trained model. The speedup thanks to pre-training ranges from 1.3x to 8x depending on the pre-trained model quality, strength of data augmentation, and dataset size. Good pre-trained models are also needed for low-data applications like PASCAL segmentation.

**The scalability, generality and flexibility of self-training.** Our experimental results highlight important advantages of self-training. First, in terms of flexibility, self-training works well in every setup that we tried: low data regime, high data regime, weak data augmentation and strong data augmentation. Self-training also is effective with different architectures (ResNet, EfficientNet, SpineNet, FPN, NAS-FPN), data sources (ImageNet, OID, PASCAL, COCO) and tasks (Object Detection, Segmentation). Secondly, in terms of generality, self-training works well even when pre-training fails but also when pre-training succeeds. In terms of scalability, self-training proves to perform well as we have more labeled data and better models. One bitter lesson in machine learning is that most methods fail when we have more labeled data or more compute or better supervised training recipes, but that does not seem to apply to self-training.

## Broader and Social Impact

Our paper studies self-training, a machine learning technique, with applications in object detection and segmentation. As a core machine learning method, self-training can enable machine learning methods to work better and with less data. So it should have broader applications in computer vision, and other fields such as speech recognition, NLP, bioinformatics etc. The datasets in our study are generic and publicly available, which do not tie to any specific application. We foresee positive impacts if the method is applied to datasets in self-driving or healthcare. But the method can also be applied to other datasets and sensitive applications that have ethical implications such as mass surveillance.

## Acknowledgements

We thank Anelia Angelova, Aravind Srinivas, and Mingxing Tan for comments and suggestions.

## Footnotes

[1]Code and checkpoints for our models are available at https://github.com/tensorflow/tpu/tree/master/models/official/detection/projects/self_training

[2]https://github.com/tensorflow/tpu/tree/master/models/official/efficientnet

[3]`https://github.com/tensorflow/tpu/tree/master/models/official/detection`

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
