[Supplementary Material]

# A  Other Related Work

Self-training is related to the method of pseudo labels [57, 66–68] and consistency training [69–88]. There are many differences between these works and ours. First, self-training is different from consistency training in that self-training uses two models (a teacher and a student) whereas consistency training uses only one model. Secondly, previous works focus on image classification, whereas our work studies object detection and segmentation. Finally, previous works do not study the interactions between self-training and pre-training under modern data augmentation.

# B  Loss Normalization Analysis

In this work we noticed that the standard loss for self-training $\hat{L} = L_h + \alpha L_p$ can be quite unstable. This is caused by the total loss magnitude drastically changing as $\alpha$ is varied. We thus implement a Loss Normalization method, which stabilizes self-training as we vary $\alpha$: $\hat{L} = \frac{1}{1+\alpha}(L_h + \alpha \frac{\bar{L}_h}{\bar{L}_p} L_p)$, where $L_h$, $L_p$, $\bar{L}_h$ and $\bar{L}_p$ are human loss, pseudo loss and their respective moving averages over training. All moving averages are an exponential moving average with a decay rate of 0.9997.

Figure 2 shows the Loss Normalization performance as we vary the data augmentation strength, training iterations, learning rate and $\alpha$. These experiments are performed on RetinaNet with a ResNet-101 backbone architecture on COCO object detection. ImageNet is used as the dataset for self-training. As can be seen from the figure, Loss Normalization gets better results in almost all settings, and more importantly, helps avoid training instability when $\alpha$ is large. Across all settings of varying augmentations, training iterations and learning rates we find Loss Normalization achieves an average of +0.4 AP performance over the standard loss combination. Importantly, it also helps in our highest performing Augment-S4 setting by +1.3 AP.

Figure 2: Performance of Loss Normalization across different data augmentation strengths, training iterations and learning rates. All results are on COCO object detection using ImageNet for self-training. The × symbol represents a run got NaNs and was unable to finish training.

Recent self-training works typically fix the $\alpha$ parameter to be one across all of their experiments [12, 87]. We find in many of our experiments that setting $\alpha$ to one is sub-optimal and that the optimal $\alpha$ changes as the training iterations and augmentation strength varies. Table 9 shows the optimal $\alpha$'s as augmentation and training iterations vary. As the augmentation strength increases the optimal $\alpha$ decreases. Additionally, as the training iterations increases the optimal $\alpha$ increases.

| Optimal Alpha | Augment-S1 | Augment-S2 | Augment-S3 |
|---|---|---|---|
| 90k Iterations | 3.0 | 2.0 | 0.5 |
| 180k Iterations | 4.0 | 3.0 | 1.0 |

Table 9: Optimal $\alpha$ as a function of augmentation strength and training iterations. For each augmentation and training iteration settings the following $\alpha$ were tried: 0.25, 0.5, 1.0, 2.0, 3.0, 4.0.

## C    Student Model Initialization Study for Self-training

| Backbone | Initialization | Augment-S1 | Augment-S2 | Augment-S3 | Augment-S4 |
|---|---|---|---|---|---|
| Eff-B7 | Random | 39.2 | 41.5 | 43.9 | 44.3 |
| Eff-B7 w/ Self-training | Teacher | 40.9 | 43.0 | 45.4 | 45.6 |
| Eff-B7 w/ Self-training | Random | 41.0 | 42.7 | 45.0 | 45.2 |

Table 10: Study on whether to initialize the student model in self-training with the teacher checkpoint or from random initialization. All models use the training methodology in Section 3.1.

In this section we study how the student model should be initialized in self-training. Table 10 shows the results of initializing the student from the teacher weights and using random initialization. Across all four augmentation regimes we observe similar performance between the two settings, with initializing from the teacher weights doing slightly better (0.3-0.4 AP). One added benefit of initializing the student with the teacher weights is not only due to the increased performance, but the speedup in convergence. Across all augmentation regimes the student model trained from scratch had to train on average 2.25 times as long as the student model initialized with the teacher weights. Therefore for all experiments in the paper we initialize the student with the teacher weights.

## D    Further Study of Augmentation, Supervised Dataset Size, and Pre-trained Model Quality

We expand upon our previous analysis in Section 4.1 and show how all four augmentation strengths across different COCO sizes interact with pre-trained checkpoint quality on COCO. Figure 3 shows the interaction of all these factors. We again observe the same three points: 1) stronger data augmentation diminishes the value of pre-training, 2) pre-training hurts performance if stronger data augmentation is used, and 3) more supervised data diminishes the value of pre-training. Across all augmentations and data sizes we also observe the better ImageNet pre-trained checkpoint, ImageNet++ , outperforming the standard ImageNet pre-trained model. Interestingly, in the three out of four augmentation regimes where pre-training hurts, the better the pre-trained checkpoint quality, the less it hurts.

Figure 3: Supervised object detection performance under various COCO dataset sizes, ImageNet pre-trained checkpoint qualities and data augmentation strengths.

As a case study in the low data regime, we study the impact of pre-trained checkpoint quality and augmentation strength on PASCAL VOC 2012. The results in Table 11 indicate that for training on

the PASCAL `train` dataset, which only has 1.5k images, checkpoint quality is very important and improves results significantly. We observe that the gain from checkpoint quality begins to diminish as the augmentation strength increases. Additionally, the performance of the ImageNet checkpoint is again correlated with the performance on PASCAL VOC.

| Setup | Augment-S1 | Augment-S4 |
|---|---|---|
| Rand Init | 28.4 | 41.5 |
| ImageNet Init | (+51.8) 80.2 | (+39.9) 81.4 |
| ImageNet++ Init | (+55.5) 83.9 | (+43.7) 85.2 |

Table 11: Supervised semantic segmentation performance on PASCAL with different ImageNet pre-trained checkpoint qualities and data augmentation strengths.

# E   ResNet-101 Self-training Performance on COCO

In the paper we presented our experimental results on COCO with RetinaNet using EfficientNet-B7 and SpineNet backbones. Self-training also works well on other backbones, such as ResNet-101 [89]. Our results are presented in Table 12. Again, self-training achieves strong improvements across all augmentation strengths.

| Setup | Augment-S1 | Augment-S2 | Augment-S3 | Augment-S4 |
|---|---|---|---|---|
| Supervised | 37.0 | 39.5 | 41.9 | 42.6 |
| Self-training w/ **ImageNet** | (+2.0) 39.0 | (+1.8) 41.3 | (+0.9) 42.8 | (+1.0) 43.6 |
| Self-training w/ **OID** | (+2.5) 39.5 | (+2.4) 41.9 | (+1.5) 43.4 | (+1.3) 43.9 |

Table 12: Performance of our four different strength augmentation policies. The supervised model is a ResNet-101 with image size $640 \times 640$ with RetinaNet using the same training protocol as EfficientNet described in 3.1 with a few minor details. Float32 instead of bfloat16 precision is used and batch norm beta/gamma are included in the weight decay regularization. This helps to improve the training stability. Also the RandAugment magntiude was increased from 10 to 15.

# F   The Effects of Unlabeled Data Sources on Self-Training

An important question raised from recent experiments is how changing the additional dataset source affects self-training performance. In our analysis we use ImageNet, a dataset designed for image classification that mostly contains *iconic* images. The image contents are known to be quite different from COCO, PASCAL, and Open Images, which contain more *non-iconic* images. Iconic images typically only have one object with its conical view, while non-iconic images capture multiple objects in a scene with their natural views [58]. Table 13 studies how changing the additional data from ImageNet to Open Images Dataset [60] impacts the performance of self-training. Switching the additional dataset source improves performance of self-training up to +0.6 AP over using ImageNet across varying data augmentation strengths on COCO. Interestingly the Open Images Dataset was found to not help COCO by pre-training in [45], but we do see improvements using it over ImageNet for self-training.

| Setup | Augment-S1 | Augment-S2 | Augment-S3 | Augment-S4 |
|---|---|---|---|---|
| Supervised | 39.2 | 41.5 | 43.9 | 44.3 |
| Self-training w/ **ImageNet** | (+1.7) 40.9 | (+1.5) 43.0 | (+1.5) 45.4 | (+1.3) 45.6 |
| Self-training w/ **OID** | (+2.0) 41.2 | (+2.1) 43.6 | (+1.6) 45.5 | (+1.7) 46.0 |

Table 13: Performance on different self-training dataset sources with varying augmentation strengths. All models use an EfficientNet-B7 backbone model on COCO object detection starting from a random initialization.

We also study the effects of changing the additional dataset source on PASCAL VOC 2012. In Table 14, we observe that changing the additional data source from ImageNet to COCO improves performance across all of our augmentation strengths. The best self-training dataset is PASCAL `aug`

set, which is in-domain data for the PASCAL task. The PASCAL `aug` set which has only about 9k images improves performance more than COCO with 240k images.

| Setup | Augment-S1 | Augment-S4 |
|---|---|---|
| Supervised | 83.9 | 85.2 |
| Self-training w/ **ImageNet** | (+1.1) 85.0 | (+0.8) 86.0 |
| Self-training w/ **COCO** | (+1.4) 85.3 | (+1.4) 86.6 |
| Self-training w/ **PASCAL**(aug set) | (+1.7) 85.6 | (+1.5) 86.7 |

Table 14: Performance on different source datasets for PASCAL Segmentation. All models are initialized using EfficientNet-B7 ImageNet++ checkpoint.

## G  Visualization of Pseudo Labels in Self-training

**PASCAL VOC dataset:**  The original PASCAL VOC 2012 dataset contains 1464 labeled in `train` set. Extra annotations are provided by [18] resulting in 10582 images in `train+aug`. Most previous works have used the `train+aug` set for training. However, we find that with strong data augmentation training with the `aug` set actually hurts performance (see Table 8). Figure 4 includes selected examples from the `aug` set. We observe the annotations in `aug` set are less accurate compared to the `train` set. For example, some of the images do not include annotation for all the objects in the image and segmentation masks are not precise. The third column of the figure shows pseudo labels generated from our teacher model. From the visualization, we observe that the pseudo labels can have more precise segmentation masks. Empirically, we find that training with this pseudo label set improves performance more than training with the human annotated `aug` set (see Table 8).

Figure 4: Human labels and pseudo labels on examples selected from PASCAL `aug` dataset. We select the examples where pseudo labels are more accurate than noisy human labels from [18].

**ImageNet dataset:**  Figure 5 shows segmentation pseudo labels generated by the teacher model on 14 randomly-selected images from ImageNet. Interestingly, some of the ImageNet classes that

don't exist in the PASCAL VOC 2012 dataset are predicted as one of its 20 classes. For instance, saw and lizard are predicted as bird. Although pseudo labels are noisy they still improve accuracy of the student model (Table 14).

Figure 5: Pseudo segmentation masks on images randomly selected from ImageNet dataset.

# H  Optimal Model Training Iterations and Alpha Weighting

In all experiments, we allow our models to train until convergence (validation set performance no longer improves). For the self-training experiments we search over a few different alpha values: [0.25, 0.5, 1.0, 2.0, 3.0] (see Appendix B for more details). Below we list all of the optimal training iterations and alphas to promote reproducibility for all of our experiments. For each table the optimal training iteration found is represented by **(45k)**, which means the model was trained for 45000 steps. The optimal alpha is represented as **(1.0)**. An alpha value of **(—)** represents that no alpha is used in the experiment as our supervised learning experiments do not make use of pseudo labeled data. One table (Table 7) jointly trains ImageNet and COCO at the same time. For this setup we simply use a scalar to combine the ImageNet loss and the COCO loss, which is represented as **(0.2)**. The total training loss is computed by $\text{Loss}_{\text{COCO}} + 0.2 \cdot \text{Loss}_{\text{ImageNet}}$.

| Setup | Augment-S1 | Augment-S2 | Augment-S3 | Augment-S4 |
|---|---|---|---|---|
| Rand Init | 39.2 **(45k) (—)** | 41.5 **(90k) (—)** | 43.9 **(90k) (—)** | 44.3 **(120k) (—)** |
| ImageNet Init | 39.5 **(22.5k) (—)** | 40.7 **(45k) (—)** | 43.2 **(90k) (—)** | 43.3 **(90k) (—)** |
| Rand Init w/ ImageNet Self-training | 40.9 **(45k) (1.0)** | 43.0 **(90k) (1.0)** | 45.4 **(90k) (0.5)** | 45.6 **(90k) (0.5)** |

Optimal training iterations and alpha for **Table 2**.

| Setup | 20% Dataset | 50% Dataset | 100% Dataset |
|---|---|---|---|
| Rand Init | 30.7 **(45k) (—)** | 39.6 **(90k) (—)** | 44.3 **(120k) (—)** |
| Rand Init w/ ImageNet Self-training | 34.1 **(90k) (3.0)** | 41.4 **(90k) (1.0)** | 45.6 **(90k) (0.5)** |
| ImageNet Init | 33.3 **(5.625k) (—)** | 38.8 **(22.5k) (—)** | 43.3 **(90k) (—)** |
| ImageNet Init w/ ImageNet Self-training | 36.0 **(90k) (3.0)** | 40.5 **(45k) (1.0)** | 44.6 **(90k) (1.0)** |
| ImageNet++ Init | 35.9 **(5.625k) (—)** | 39.9 **(11.25k) (—)** | 43.8 **(45k) (—)** |
| ImageNet++ Init w/ ImageNet Self-training | 37.2 **(90k) (3.0)** | 41.5 **(45k) (1.0)** | 44.6 **(45k) (0.25)** |

Optimal training iterations and alpha for **Table 3**.

| Setup | COCO AP |
|---|---|
| Rand Init | 41.1 **(200k) (—)** |
| ImageNet Init (Supervised) | 40.4 **(160k) (—)** |
| ImageNet Init (SimCLR) | 40.4 **(160k) (—)** |
| Rand Init w/ Self-training | 41.9 **(120k) (0.25)** |

Optimal training iterations and alpha for **Table 4**.

| Model | # FLOPs | # Params | AP (`val`) | AP (`test-dev`) |
|---|---|---|---|---|
| SpineNet-143[†] (1280) | 524B | 67M | 50.9 **(472k) (—)** | 51.0 |
| SpineNet-143 (1280) w/ Self-training | 524B | 67M | **52.4 (472k) (0.25)** | **52.6** |
| SpineNet-190[†] (1280) | 1885B | 164M | 52.6 **(370k) (—)** | 52.8 |
| SpineNet-190 (1280) w/ Self-training | 1885B | 164M | **54.2 (370k) (0.5)** | **54.3** |

Optimal training iterations and alpha for **Table 5**. Note due to the high computational demands of this experiment only a subset of alphas and training iterations were tried. For SpineNet-143 alphas of (0.25, 0.5, 1.0) were tried and only a single training iteration. For SpineNet-190 only a single alpha and training iteration were tried.

| Model | Pre-trained | # FLOPs | # Params | mIOU (`val`) | mIOU (`test`) |
|---|---|---|---|---|---|
| Eff-B7 | ImageNet++ | 60B | 71M | 85.2 **(40k) (1.0)** | — |
| Eff-B7 w/ Self-training | ImageNet++ | 60B | 71M | **86.7 (40k) (1.0)** | — |
| Eff-L2 | ImageNet++ | 229B | 485M | 88.7 **(20k) (1.0)** | — |
| Eff-L2 w/ Self-training | ImageNet++ | 229B | 485M | **90.0 (20k) (1.0)** | **90.5** |

Optimal training iterations and alpha for **Table 6**.

| Setup | Sup. Training | w/ Self-training | w/ Joint Training | w/ Self-training w/ Joint Training |
|---|---|---|---|---|
| Rand Init | 30.7 **(45k) (—) (—)** | 34.1 **(90k) (3.0) (—)** | 33.6 **(45k) (—) (0.5)** | 35.1 **(90k) (2.0) (0.2)** |
| ImageNet Init | 33.3 **(5.625k) (—) (—)** | 36.0 **(90k) (3.0) (—)** | 34.0 **(90k) (—) (0.5)** | 36.6 **(90k) (2.0) (0.2)** |

Optimal training iterations, alpha and ImageNet loss weighting for **Table 7**.

| Setup | train | train + aug | train + aug w/ Self-training |
|---|---|---|---|
| ImageNet Init w/ Augment-S1 | 83.9 (40k) (1.0) | 84.7 (40k) (1.0) | 85.6 (40k) (1.0) |
| ImageNet Init w/ Augment-S4 | 85.2 (40k) (1.0) | 84.8 (40k) (1.0) | 86.7 (40k) (1.0) |

Optimal training iterations and alpha for **Table 8**.

| Backbone | Initialization | Augment-S1 | Augment-S2 | Augment-S3 | Augment-S4 |
|---|---|---|---|---|---|
| EfficientNet-B7 | Random | 39.2 | 41.5 | 43.9 | 44.3 |
| EfficientNet-B7 w/ Self-training | Teacher | (45k) 40.9 | (90k) 43.0 | (90k) 45.4 | (90k) 45.6 |
| EfficientNet-B7 w/ Self-training | Random | (180k) 41.0 | (135k) 42.7 | (180k) 45.0 | (135k) 45.2 |

Optimal training iterations and alpha for **Table 10**.

| Setup | Augment-S1 | Augment-S2 | Augment-S3 | Augment-S4 |
|---|---|---|---|---|
| Supervised | 37.0 (45k) (—) | 39.5 (90k) (—) | 41.9 (90k) (—) | 42.6 (180k) (—) |
| Self-training w/ **ImageNet** | 39.0 (90k) (1.0) | 41.3 (90k) (1.0) | 42.8 (90k) (0.5) | 43.6 (180k) (0.25) |
| Self-training w/ **OID** | 39.5 (90k) (2.0) | 41.9 (90k) (2.0) | 43.4 (90k) (0.5) | 43.9 (180k) (0.5) |

Optimal training iterations and alpha for **Table 12**.

| Setup | Augment-S1 | Augment-S2 | Augment-S3 | Augment-S4 |
|---|---|---|---|---|
| Supervised | 39.2 (45k) (—) | 41.5 (90k) (—) | 43.9 (90k) (—) | 44.3 (120k) (—) |
| Self-training w/ **ImageNet** | 40.9 (45k) (1.0) | 43.0 (90k) (1.0) | 45.4 (90k) (0.5) | 45.6 (90k) (0.5) |
| Self-training w/ **OID** | 41.2 (90k) (3.0) | 43.6 (90k) (2.0) | 45.5 (90k) (0.5) | 46.0 (120k) (0.5) |

Optimal training iterations, alpha and ImageNet loss weighting for **Table 13**.

| Setup | Augment-S1 | Augment-S4 |
|---|---|---|
| Supervised | 83.9 | 85.2 |
| Self-training w/ **ImageNet** | 85.0 (40k) (1.0) | 86.0 (40k) (1.0) |
| Self-training w/ **COCO** | 85.3 (40k) (1.0) | 86.6 (40k) (1.0) |
| Self-training w/ **PASCAL**(aug set) | 85.6 (40k) (1.0) | 86.7 (40k) (1.0) |

Optimal training iterations and alpha for **Table 14**.