[Reviews · NeurIPS 2020]

Review 1

Summary and Contributions: This paper works on a very interesting an important regime of machine learning and specifically computer vision. Properly using myriads of unlabeled data to aid supervised learning has been in the radar of many machine learning researchers. This paper presents a methodical treatise on different sources for imparting knowledge from unlabeled (self-training, self-supervision) or non-targeted labeled data (pretraining). Self-training is shown to work good where the advantage of pretraining is not apparent. The authors took detection and segmentation as the use cases and showed that self-training is always a boon irrespective of whether pretraining helps [the case with small amount of labeled data for supervised training] or not [when amount of labeled data for supervised training is present at large]. The work builds on top of noisy student [10] and studies the benefit of noisy student training over pretraining with different data augmentation strategies across varied sizes of labeled data. The contribution of the work is in showing that (with very few practical limitations e.g., self-training data and supervised training data being not widely differing in nature) self-training can be useful in two completely opposing scenarios: It is a method that works for both 1. when labeled data is hard to get and when 2. labeled data is not so difficult to get.

Strengths: 1. The work addresses a very interesting, relevant and important problem to the machine learning community. Learning from only a few labeled images is helped by initializing with weights of models pretrained on already existing large labeled dataset that may not necessarily have been collected for the same task as the one for which the few labeled data is available. Another helping hand is to use self-training where unlabeled data is used cleverly to get free and plentiful supervisory signal. Yet another regime is self-training where an already strong model trained on large annotated dataset gives rise to a teacher for generating reasonable automated pseudo-labels for the unlabeled data to be used as labeled training data. The paper, in question, performs an extensive comparison on these three approaches and sealed the superiority of self-training over the other two approaches on a large spectrum of control experiments. 2. Experiments is the major strength of the paper. And by ‘experiments’ I don’t mean ‘brute-force’ experiments only. Starting from designing and carrying out the experiments to analysis of the same is done very good. One specific example is the motivation to run the experiments in section 4.3. The motivation comes from the combination of two facts: 1. In highest data and strongest augmentation setting supervised pretraining hurts (ref Table 3) but self-training with the same supervised dataset (treated as unlabeled data) helps 2. Self-supervised pretraining also does not use label information. Since discarding label information in self-training helped, could it be the case that discarding label information and using self-supervised pretraining will also help? The experimental results though do not support the hypothesis. This, however, was worth trying and a necessary experiment to do to rubber stamp the authority of noisy self-training. The ablation, comparison and analysis of the results are well performed.

Weaknesses: Though the paper is strong in terms of experiments and methodologies/intuitions a few clarifications on the following would certainly help. 1. The paper builds on the recent noisy student paper [10] which also shows its usefulness on robustness tests. Is there any comparison between supervised pretraining, self-supervised pretraining and self-training on robustness analysis? 2. Why are the COCO AP values in table 4 last row and table 3 second row (Rand Init w/ ImageNet Self-training) different? 3. This is regarding Fig. 5 of the supplementary. Are these images all used as the pseudolabeled images for the student training? For example, the second row, first image gives all background pseudo labels. Is this image used in self-training the student?

Correctness: The experimental results are supportive of the claims, as far as I can see. One specific example is - In the abstract the claim that was made about self-training working well in the exact same setup where pretraining does not work, is well supported by the results shown in table 2,3,5 and 6.

Clarity: The paper is generally written well with requisite motivation, hypothesis description and analysis. The figures and tables are self-sufficient with descriptive captions and in-section texts. The introduction provides brief summary, contributions and main findings to keep up reader’s interest. The authors did a good job in describing the control factors and experimental settings in detail along with the major intuition behind each experiment which made the later flow smooth and increased the readability of the experiments section.

Relation to Prior Work: Due to a long standing interest, prior literature in self and semi supervised setting is quite huge. The authors did a good job in briefly mentioning the major, important and recent works. The authors also differentiated their work with the most close work [10] in appendix A by stating clearly the difference in scope of the proposed work in studying the interactions between self-training and pre-training under modern data augmentation.

Reproducibility: Yes

Additional Feedback: The volume of the experiments, experiment design and analysis are major strengths of the paper. However, a few clarifications as listed in point 3 can help improve the paper even further. =====After Rebuttal===== Had a careful read of the rebuttal and fellow reviewers’ comments. My concern was adequately addressed. I see that most concerns from all the reviewers are coming from the future scope of the approach – e.g., tackling robustness (from me), instance segmentation, fine-grained recognition, open set embedding (R2, R4) etc. The responses to these are quite adequate and insightful. With such experiments and capability to instigate future research question, I think, if published, this paper can motivate many other related tracks. So, I am going for an accept.


Review 2

Summary and Contributions: Taking mainstream computer vision tasks such as object detection and semantic segmentation as the test cases, this paper studies the impact of self-training under diverse controlling factors such as different data augmentation methods, data sizes and network capacities. Under these controlling factors, the authors compare the performance of self-training with widely used pre-training, and also self-supervised learning, with comprehensive experiments on popular datasets including ImageNet, OpenImages, MS COCO and PASCAL VOC.

Strengths: + The paper is well written. + Comprehensive experiments on large scale visual recognition datasets. + Experiments need huge computational resources. + Some interesting conclusions.

Weaknesses: Generally, this paper is purely experiments-driven. Although it sheds some new insights/observations on the use of self-training, pre-training and self-supervised learning under different controlling factors such as different data augmentation methods, data sizes and network capacities, the theoretical contribution of the paper is limited. Besides, the related analysis on why self-training works better than pre-training and self-supervised learning is not sufficient, or say the reason is not clear enough. One critical question may need more analysis is the benefit of using data augmentation (especially strong ones such as AutoAugment and RandAugment) vs. training strategy like self-training. Which one is more important to final performance improvement? For experiments on MS COCO, how about the result differences on instance segmentation? Are the conclusions similar to those on object detection and semantic segmentation? I am wondering, when applying the same experiment setups to fine-grained object recognition tasks like face recognition etc., similar conclusions still hold? In many computer vision tasks, for real deployment, much smaller neural networks are needed due to constrained compute resources. Do similar conclusions still hold when using low-capacity models? Especially when data augmentation shows relatively small gain to final models, how about the performance of self-training works vs. pre-training and self-supervised learning? Self-training plus data augmentation leads to heavy increase in training cost. Pose rebuttal ----------------------------------------------------------------------------------------------------------- Most of my concerns are well addressed by the author's feedback, thus I increase my rating from 6 to 7. Generally, this is a decent paper. It will be better to add some results and analysis on fine-grained object recognition tasks to final version.

Correctness: Mostly yes.

Clarity: Yes.

Relation to Prior Work: Pretty good.

Reproducibility: Yes

Additional Feedback: Please see my detailed comments to weaknesses.


Review 3

Summary and Contributions: The paper investigates the effect of ImageNet pre-training, self-supervised pre-training and self-training on downstream task performance. Surprisingly, with strong augmentations, initializing with ImageNet pre-trained weights is worse than training from scratch. This effect also occurs when initializing with a self-supervised method. Finally, self-training (labelling images with a model trained on a smaller dataset) consistently improves the results in all settings. The paper evaluates two downstream tasks: object detection and semantic segmentation on several combinations of datasets with consistent results.

Strengths: S1. Understanding the effect of pre-training and self-training in various CV tasks is important since the common practice at the moment is almost always using ImageNet pre-training. S2. The paper contains very interesting findings. Self-training seems to be consistently better than pre-training - at least for the tasks of object detection and semantic segmentation. S3. The paper is well written, and presents extensive empirical evidence for all claims.

Weaknesses: W1. Apart from the insight that ImageNet pre-training can hurt performance and that self-training is beneficial, there is little understanding gained. It is not clear why this is the case. Understanding this would be very impactful (especially for self-supervised learning) since it is commonly believed that these methods learn quite general representations. It is of course very difficult to answer this question. However, one could think of some experiments that shed further light: To rule out some kind of distribution shift between datasets, one could compare self-training, random-init and self-supervised pre-training on the same dataset by splitting it in half. This would guarantee that the observed effect cannot be attributed to a change in image distribution between pre-training and final training. Another interesting experiment to show more insights could be the reverse setup: pre-training on COCO and then fine-tuning on ImageNet. Would that show a similar behaviour, or is the classification task too restrictive to capture enough information for detection but not vice-versa? W2. As stated in W1 it is quite surprising that SimCLR init hurts COCO performance as much as ImageNet init (Tab. 4). To better understand the interaction with self-training, one could add SimCLR with self-training to the table. This would show whether self-supervised initialization is better than random initialization when combined with self-training. W3. (minor) the PascalVOC results in Tab.6 are difficult to compare to the state of the art since different back-bones architectures have been used. A comparison using DeepLabv3 could help to show a clearer picture. +++++++ After Author Feedback +++++++ After reading the other reviews and the rebuttal, I find most of my concerns sufficiently addressed and my views aligning with the other reviewers. The paper has discovered an interesting observation: that self-training seems to be preferable over pre-training under certain augmentation strategies. This is a good starting point for further investigation and should be published.

Correctness: The claims are evaluated thoroughly through a series of experiments on different tasks (Segmentation & Detection) on several combinations of datasets (COCO, ImageNet, OpenImages, PascalVOC).

Clarity: The paper is well written and easy to follow. The chosen structure highlights all findings and their empirical validation sequentially which is good to follow and does not require jumping between pages.

Relation to Prior Work: The paper is well positioned in relation to existing work. The difference to the closest work [1] is clear as the submission extends the analysis to self-supervised pre-training and self-training.

Reproducibility: Yes

Additional Feedback: Minor comments that did not influence my rating - The NeurIPS style guide suggests the captions for tables to be above the table and not below.


Review 4

Summary and Contributions: This paper investigates self-training methods for utilizing additional data and compares self-training against supervised/self-supervised pre-training methods. Several important conclusions are given: First, pre-training helps in low-data/weak augmentation cases but hurts in high-data/strong-augmentation cases. Second, self-training shows better performance than pre-training in both low-data and high-data regimes; Last, self-training is additive to pre-training.

Strengths: This is a typical Google-style paper: The authors attempt to show us some simple but interesting results and perform extensive experiments to support their findings. In general, this is a solid paper. The writing is good, experiments are sufficient to make the authors' points, and the final performance is surprisingly high. The fact that self-training is additive to pre-training seems very helpful.

Weaknesses: Below are my concerns, 1. My biggest concern is that this paper only tells us some conclusions but does not provide in-depth interpretations of these conclusions, although the conclusions are useful and well-validated by experiments. For instance, the underlying reason why self-annotated labels are better than noisy human-annotated labels (Table. 8) is still not clear. The authors' explanation that the pseudo labels are more "targeted" is not so convincing. Another example is in Section 5, "The benefit of joint-training", the authors show an interesting finding that self-training is also additive to joint-training, but the underlying reasons are not explored. I believe this paper would be more rounded if some of such questions are answered properly. 2. It seems that the self-training method merely applies to close-set tasks such as classification (Object detection and semantic segmentation can be formulated as classification tasks). If the target task is an open-set task, such as face recognition, the self-training method cannot be directly adopted. In real-world applications, many tasks are open-set, so that the application scope of self-training is not as wide as supervised/self-supervised pre-training. 3. Regarding experiments: In L#223, the authors describe that the results are obtained using 100 COCO data and S-4 augmentation. In Table 4, the COCO AP of Rand Init + 100% data + S-4 augmentation is 41.1, but in Table 3, this number is 44.3. These numbers are a bit confusing. Post rebuttal: I have read the author feedback and comments from other reviewers. Thanks the authors for their careful explain and most of my concerns are addressed. It is interesting to see the self-training method can also apply to open-set tasks such as face recognition (by annotating pair-wise relationship as mentioned in authors' response), and I believe self-training would be very helpful for the community, so I maintain my rating as 'accept'.

Correctness: Yes

Clarity: Yes

Relation to Prior Work: Yes

Reproducibility: Yes

Additional Feedback:

[Author Response · NeurIPS 2020]

(R1) **Robustness analysis for supervised pre-training, self-supervised pre-training, and self-training.** We agree
that robustness for pre-training and self-training is an interesting direction. However, unlike ImageNet, there is no
robustness benchmark for detection and segmentation. This can be very interesting future follow-up work.

(R1, R4) **The performance in Table 3 and Table 4 does not match.** The backbone models are not the same in Table
3 (EfficientNet-B7) and Table 4 (ResNet-50), because SimCLR does not have an EfficientNet checkpoint. We will
revise and make it more clear in the caption.

(R1) **Are the images in Fig. 5 all used as the pseudo-labeled images for the student training?.** Yes, that's correct.
We do not filter any images even if they only contain the background class.

(R2, R3, R4) **Analysis of why self-training performs better than pre-training.** Our discussion section attempted to
address our two hypotheses on why self-training outperforms pre-training. We will update the section to more clearly
address the question. **Hypothesis 1**): Joint optimization of human and pseudo labeled data; **2**): Task alignment.

**Joint Optimization (1)**: In Table 7 we study the benefits of joint optimization with two experiments: jointly training
ImageNet classification and COCO object detection and pre-training w/ ImageNet classification and fine-tuning on
COCO. Our results reveal that jointly training the ImageNet objective with COCO is more effective than pretraining a
model with the ImageNet objective and then fine-tuning on COCO.

**Task Alignment (2)**: Table 8 studies the importance of task alignment. Pascal consists of two parts: a standard train
set ('`train`') and a train set labeled with a different distribution of human annotations ('`trainaug`'). We observe
training a teacher on '`train`' then relabeling '`trainaug`' outperforms using the original annotations of '`trainaug`'
when training on both datasets concurrently (84.8 *vs.* 86.7 mIoU). We observe using targeted pseudo labels is more
useful than using ground truth human labels that do not well match the target labels.

(R2) **Which is more important: strong data augmentation or self-training?** **(1)** Self-training is quite additive
to data augmentation and **(2)** self-training is more general than data augmentation methods as it does not require
domain knowledge. In Table 2 the best data augmentation yields 5.1AP improvement and self-training over a +1.3
AP improvement across all augmentation methods. Therefore we do not see a tradeoff between the two methods and
suggest using both data augmentation and self-training. Furthermore, self-training can also be done without any dataset
knowledge, where data augmentation methods have to be crafted according to the task at hand. If we apply self-training
to a self-driving car dataset, the data augmentation method needs to change whereas self-training can stay the same.

(R2) **Will instance segmentation on COCO show similar conclusions as box detection and semantic segmenta-**
**tion?** Our hypothesis is yes. Unfortunately, we do not have the experiments to answer the question.

(R2, R4) **Will the conclusions be similar to fine-grained recognition problem or open-set embedding learning**
**problem, e.g., face recognition?** Great question! We want to argue that it is possible to apply self-training to open-set
recognition (such as face recognition), where the training labels are typical defined by similar/dissimilar pairs of
examples. The teacher model can also label similar and dissimilar examples in the unlabelled dataset for open-set tasks.

For fine-grained/open-set recognition problem, there can be more noise in the pseudo labels. We observe self-training
can work well even when the pseudo labels are noisy. For example, Figure 5 shows several wrong pseudo segmentation
labels (e.g., mislabel a saw with the bird) because the concept does not exist in the teacher model. Nevertheless, Table
13 shows improvements using these examples for self-training. This hints that self-training has a certain degree of
robustness against noise in pseudo labels caused from domain or label space shift.

(R2) **Does self-training benefit low capacity models?** We experiment with a wide variety of different model capacities
in our paper:ResNet-50, ResNet-101, EfficientNet-B7, and EfficientNet-L2. All these models show consistent benefits
when applying self-training. However, we do not experiment with mobile size models such as MobileNet. In the
classificiation domain [12] applied self-training to a mobile sized model and sees good improvements.

(R3) **Additional experiments to control the domain shift, swap the labeled and unlabeled datasets, and use**
**SimCLR checkpoint for self-training.** Thanks for suggesting extra experiments. Due to the limitation of time and
page limit, we are not able to try all these ideas. Chen *et al.* [a] shows promising results combining self-supervised
training and self-training.

(R3) **Comparison of DeepLab and our model.** Here we want to show that self-training can work for semantic
segmentation in a high performance regime by using the EfficientNet + FPN architecture. Therefore having a baseline
EfficientNet with and without self-training proves our point.

[a] Chen, T., Kornblith, S., Swersky, K., Norouzi, M., Hinton, G.E. (2020). Big Self-Supervised Models are Strong
Semi-Supervised Learners. ArXiv, abs/2006.10029.


[Meta-Review · NeurIPS 2020]

All four reviewers unanimously voted for acceptance. They appreciated that the paper addresses an important and timely question (When is self-training beneficial over pre-training?) and numerous experiments and analyses provide novel insights that will encourage future research in this direction. The initial reviews pointed out a few (minor) concerns and the rebuttal adequately addressed them. In response, R2 increased the rating from 6 to 7 after the discussion phase. Overall, I think this is a solid paper and would love to see it at the conference.